# Modulating Glycolysis to Improve Cancer Therapy

**DOI:** 10.3390/ijms24032606

**Published:** 2023-01-30

**Authors:** Chaithanya Chelakkot, Vipin Shankar Chelakkot, Youngkee Shin, Kyoung Song

**Affiliations:** 1Bio-MAX/N-Bio, Seoul National University, Seoul 08826, Republic of Korea; 2Department of Cancer Biology, Lerner Research Institute, Cleveland Clinic, Cleveland, OH 44195, USA; 3Laboratory of Molecular Pathology and Cancer Genomics, Research Institute of Pharmaceutical Science, Department of Pharmacy, Seoul National University, Seoul 08826, Republic of Korea; 4Department of Molecular Medicine and Biopharmaceutical Sciences, Graduate School of Convergence Science and Technology, Seoul National University, Seoul 08826, Republic of Korea; 5College of Pharmacy, Duksung Women’s University, Seoul 01366, Republic of Korea

**Keywords:** glycolysis, cancer metabolism, combination therapy

## Abstract

Cancer cells undergo metabolic reprogramming and switch to a ‘glycolysis-dominant’ metabolic profile to promote their survival and meet their requirements for energy and macromolecules. This phenomenon, also known as the ‘Warburg effect,’ provides a survival advantage to the cancer cells and make the tumor environment more pro-cancerous. Additionally, the increased glycolytic dependence also promotes chemo/radio resistance. A similar switch to a glycolytic metabolic profile is also shown by the immune cells in the tumor microenvironment, inducing a competition between the cancer cells and the tumor-infiltrating cells over nutrients. Several recent studies have shown that targeting the enhanced glycolysis in cancer cells is a promising strategy to make them more susceptible to treatment with other conventional treatment modalities, including chemotherapy, radiotherapy, hormonal therapy, immunotherapy, and photodynamic therapy. Although several targeting strategies have been developed and several of them are in different stages of pre-clinical and clinical evaluation, there is still a lack of effective strategies to specifically target cancer cell glycolysis to improve treatment efficacy. Herein, we have reviewed our current understanding of the role of metabolic reprogramming in cancer cells and how targeting this phenomenon could be a potential strategy to improve the efficacy of conventional cancer therapy.

## 1. Introduction

Cancer cells reprogram their metabolism to promote growth, metastasis, and survival. They exhibit an increased glycolytic dependency and show an elevated glucose uptake and fermentation of glucose to lactate to meet the heightened anabolic needs for cancer cell proliferation [1]. Increased glycolysis is not only important for meeting the energy needs of the cells but is also crucial for the generation of metabolic intermediates necessary for macromolecule synthesis in cancer cells [2,3]. This phenomenon, often referred to as the ‘Warburg effect,’ is observed even in the presence of completely functional mitochondria [4]. The Warburg effect has been studied for over 90 years, and several studies have explored the mechanisms governing the increased glycolytic dependency of cancer cells. Several oncogenic proteins and tumor suppressors, including the hypoxia-inducible factor (HIF-1), Myc, p53, and PI3K/Akt/mTOR pathway, have been implicated in regulating this cancer cell-specific metabolic reprogramming.

The altered metabolism in the cancer cells provides an avenue for developing cancer cell-specific therapeutic targets and anti-cancer agents. Indeed, therapeutic strategies that target glycolysis and cancer cell-specific biosynthetic pathways are a major focus area in cancer research. Although the increased glycolytic dependency of neoplastic cells suggests the potential therapeutic efficacy of glycolytic inhibitors in cancer treatment, glycolytic inhibition alone is not effective in a clinical setting [5]. Targeting metabolism, especially in combination with chemotherapy, is expected to improve therapy responses and may help overcome drug resistance [6]. Elevated glycolysis in cancer cells and the resulting lactic acidosis modulate the tumor stroma to a pro-tumorigenic microenvironment. Targeting glycolytic changes in the tumor microenvironment has been shown to be a safe and effective strategy to enhance therapeutic efficacy [7,8]. In this review, we explore and discuss glycolytic modulation in cancer cells and how it could aid as a therapeutic strategy in combination therapies with chemotherapy and radiotherapy, immunotherapy, hormonal therapy, and photodynamic therapy (PDT).

## 2. Modulating Glycolysis to Improve Chemotherapy and Radiotherapy

Metabolic modulation has been shown to sensitize cancer cells toward chemotherapy and radiotherapy. Increased glycolysis, facilitated by an increased glucose uptake, is the major energy source in cancer cells apart from being the major source of macromolecules for cell proliferation and survival [9]. Recently it was demonstrated that glycolysis-addicted cancer cells show metabolic rewiring via mTORC1 activation [10,11]. Sustained mTORC1 activation bypasses glycolysis by directing the glucose flux toward the pentose phosphate pathway. Metabolic rewiring, including dysregulated glycolysis, elevated ATP production, and cell-death escape mechanisms, are the major culprits for therapeutic resistance in cancer cells. The intracellular ATP level in cancer cells is also associated with metastasis and stemness. Thus, targeting glycolysis or intracellular/extracellular ATP levels [12,13,14] is a promising strategy to sensitize cancer cells toward chemotherapy. Several studies have reported that glycolytic inhibitors improve the efficacy of cancer treatment and that glycolytic inhibition is a promising strategy when used as a combination therapy with other treatment modalities. In line with this, inhibiting glycolytic enzymes, hexokinase (HK) [15], pyruvate kinase (PK) [16], and lactate dehydrogenase (LDH) [17], have shown sensitizing effects with several chemotherapeutic agents [5].

### 2.1. Targeting Glucose Transporters and Glucose Uptake to Improve Chemotherapy

The inhibition of glucose transporters (GLUTs), a critical rate-limiting step in glucose metabolism, modulates the therapeutic efficacy of several drugs. GLUT expression is elevated in several types of cancers and is associated with a poor prognosis suggesting their key role in cancer cell metabolism [18,19,20,21,22]. Glycolysis inhibition using inhibitors of GLUT1, when combined with routine cancer therapy, has proven to be relevant in potentiating their effects in a synergistic manner in pre-clinical studies for several cancers [19,23,24,25,26]. GLUT1 inhibition curbs the self-renewing capacity and tumor-initiating potential of cancer stem cells and has a substantial significance from a therapeutic perspective [27].

A widely studied small molecule inhibitor of GLUT1, WZB117 synergistically inhibits breast cancer cells by inducing DNA damage when treated in combination with an allosteric AKT inhibitor [24,28]. WZB117 also sensitizes breast cancer cells toward treatment with adriamycin [29] and radioresistant breast cancer cells to radiotherapy [27,30]. Previous studies have reported the synergistic effects of GLUT1 inhibitor #43 in melanoma cells and have shown that GLUT1 inhibition induces apoptosis, intracellular reactive oxygen species (ROS) generation, and the loss of mitochondrial membrane potential. Combination therapy with GLUT1 inhibitor #43 enhances the DNA-damaging effects of cisplatin by regulating the AKT/mTOR pathways [24]. Another GLUT1 inhibitor, BAY-876, enhances the cisplatin-mediated inhibition of esophageal squamous cell carcinoma [23]. siRNA-mediated GLUT1 inhibition also showed similar results and improved the efficacy of low-dose cisplatin treatment [23]. In vivo studies in uterine cancer, patient-derived models have shown that glycolytic activation contributes to the stemness of uterine endometrial cancer, and BAY-876-mediated GLUT inhibition synergistically suppressed endometrial cancer cell proliferation when used in combination with paclitaxel [31].

Combination therapy with GLUT modulators can also improve the bioavailability of chemotherapeutic drugs. The co-treatment of paclitaxel with silybin (a GLUT modulator) significantly improved the oral bioavailability of the drug in several in vitro and in vivo studies and overcame the major drawback of the limited oral bioavailability of paclitaxel [32,33]. A nanomedicine-based combination therapy using GLUT1 inhibitor and chemotherapeutic agent, curcumin, deprived cancer cells of glucose and sensitized cancer cells to chemotherapy, induced apoptosis, improved anti-tumor effects, and alleviated side-effects in vitro and in vivo [34]. Thus, combination therapy with GLUT1 inhibitors might be a rational therapeutic strategy and could also allow for low-dose treatment with chemotherapy drugs providing a paradigm for high-efficacy, low-toxicity therapeutic options [35].

Another straightforward and interesting strategy to improve the effectiveness of chemotherapy is to deprive cancer cells of glucose [36]. Icard et al. proposed that the modulation of glucose intake in combination with chemotherapy could improve the efficacy of the drug via the deprivation of ATP to cancer cells. A recent study showed that intermittent fasting throughout chemotherapy was well tolerated in patients and reduced chemotherapy-induced toxicity as measured by hematologic, metabolic, and inflammatory parameters [37]. Contrastingly, an opposite effect was reported in a pre-clinical model of pancreatic ductal adenocarcinoma (PDAC), wherein a relative glucose abundance sensitized PDAC cells to chemotherapy. Hyperglycemic patients with stage IV PDAC showed an enhanced response to chemotherapy, possibly via impaired glutathione biosynthesis [38]. A case report on non-small cell lung cancer (NSCLC) with bone and brain metastasis reported the efficacy of glucose uptake inhibition in combination with chemotherapy as a palliative treatment strategy. Fasting-induced hypoglycemia or insulin-induced hypoglycemia combined with low-dose chemotherapy could benefit cancer patients, particularly those who do not tolerate the conventional dosage of drugs [39].

### 2.2. Targeting Glycolysis Enzymes to Improve Chemotherapy and Radiotherapy

The enhanced glycolysis in the cancer cells correlates with an upregulation and activation of critical glycolytic enzymes. Targeting the key glycolytic enzymes is a promising strategy to rewire the altered tumor metabolism, to sensitize (or re-sensitize, when resistance develops) cancers to chemotherapy.

HK catalyzes the first step in glucose metabolism and converts glucose to glucose-6-phosphate. Four HK isoforms, HK1-4, with different cellular distributions and glucose affinity have been identified. HK1 and HK2 are located on the outer mitochondrial membrane and are associated with AKT-mediated cell survival [40,41]. Further, HK2 is associated with the recurrence and poor prognosis of breast cancer (BC) [42]. HK2 expression is also elevated in lung cancer, and shows significant association with the tumor stage. The deletion of the Hk2 gene in lung cancer cells ameliorated glucose-derived ribonucleotides and glutamine-derived carbon utilization in anaplerosis [43]. Targeting HK2 inhibits cell proliferation and shifts the metabolic profile of cancer cells from glycolytic to oxidative phosphorylation (OXPHOS) [43,44].

2-Deoxy glucose (2-DG), an HK inhibitor, has been shown to sensitize cancer cells to chemotherapy and radiotherapy and is being investigated in clinical trials. 2-DG is a glucose analog that triggers the intracellular accumulation of 2-deoxy-d-glucose-6-phosphate (2-DG6P), inhibiting the function of HK and glucose-6-phosphate isomerase [45]. Glycolysis inhibition with 2-DG can improve the therapeutic efficacy of trastuzumab in treating HER2+ BC [46]. Similarly, the therapeutic efficacy of paclitaxel was enhanced when treated in combination with 2-DG in in vivo studies in NSCLC and osteosarcoma models [47]. A recent study showed that 2-DG could sensitize glioblastoma cells to chloroethyl nitrosourea by regulating glycolysis, intracellular ROS generation, and endoplasmic reticulum stress induction [48]. Another study reported that combining 2-DG with autophagy inhibiting drug hydroxychloroquine enhances apoptosis in BC cells. The inhibition of autophagy combined with 2-DG induced the accumulation of misfolded proteins in the endoplasmic reticulum and resulted in sustained endoplasmic reticulum stress, induced through the pERK-eIF2α-ATF4-CHOP axis to enhance apoptosis in BC cells [49]. Although a few clinical trials (NCT05314933, NCT00096707) are investigating the toxicity, tolerability, pharmacokinetics, and recommended dose of 2-DG in advanced tumors [50], additional studies and trials are required to characterize the mechanism of action and treatment benefits of 2-DG in cancer.

Curcumin, another compound with a proven anti-tumor effect, is also known to inhibit HK expression by inhibiting transcriptional repressor SLUG. Combination therapy of curcumin with docetaxel demonstrated a high response rate, low-toxicity, and improved patient tolerance in prostate cancer [51].

3-Bromopyruvate acid (3BrPA) is another classic glycolytic inhibitor, which inhibits several enzymes in the glycolytic pathway, including HK and LDH, and is a potent inhibitor of cancer cell growth [52,53,54,55]. Combining 3BrPA with rapamycin enhanced the anti-tumor efficacy through the dual inhibition of mTOR signaling and glycolysis in LC and neuroblastoma [56,57]. In BC cells, 3BrPA enhanced the expression of thioredoxin interacting protein (TXNIP) and inhibited HK2 expression via c-Myc downregulation [58], and enhanced tamoxifen-induced cytotoxicity in vitro [59]. The combination regimen with tamoxifen also enhanced oxidative stress and reduced glutathione levels in cells, and affected tumor angiogenesis and metastasis in animal models [59]. Further, the intra-cranial delivery of 3BrPA with temozolomide showed synergistic effects and increased survival in animal models of glioma [60]. Moreover, enhanced therapeutic efficacy was demonstrated when 3BrPA was combined with sorafenib in murine models of liver cancer [61]. 3BrPA can also enhance the anti-tumor effect of low-dose radiation via the reprogramming of mitochondrial metabolism and hindering of ATP generation [62]. 3BrPA also inhibits monocarboxylate transporter 1 (MCT1) expression, which mediates the bidirectional transport of lactate in cancer cells, and sensitizes cancer cells to ionizing radiation [63].

Elevated glycolysis in cancer cells results in the conversion of pyruvate to lactate, even under aerobic conditions. Lactate is excreted at high levels from tumor cells and acts as a metabolic fuel and oncometabolite with signaling properties. Lactate utilization by tumor cells depends on the expression of monocarboxylic transporters (MCTs), which are upregulated in cancer cells [64,65]. MCTs facilitate the shuttle of lactate from cancer cells to neighboring cells, tumor stroma, and tumor-associated endothelial cells and induce metabolic rewiring. Lactate is involved in several tumorigenic functions of cancer cells, including tumor microenvironment modulation and tumor angiogenesis [66]. Targeting MCTs has been shown to suppress tumor growth in cancer cells [67]. MCT1 and MCT4 inhibitors impair leukemia cell proliferation and enhance their sensitivity toward chemotherapy [68]. Currently, a phase 1 clinical trial is evaluating the toxicity and pharmacokinetic profile of AZD3965, an MCT inhibitor in cancer therapy for B-cell lymphoma [69,70].

LDH is a key glycolytic enzyme that is elevated in aggressive cancers and is essential for tumor maintenance [71,72,73]. LDH-A is regulated by numerous oncogenic transcription factors, including c-Myc and HIF-1, and is closely associated with malignant phenotypes of cancer cells [74]. LDH-A overexpression upregulated AKT phosphorylation and PI3K, which upregulated cyclin D1 and c-Myc expression in LC cells [75,76,77,78]. LDH overexpression is also associated with epithelial–mesenchymal transition (EMT)-related genes, SNAIL and SLUG, and is thus involved in regulating the metastatic progression of cancer cells [79]. LDH-A levels could thus serve as a biomarker for cancer diagnosis and prognosis [80,81,82]. LDH inhibition induces oxidative stress, impacting cancer stem cells’ renewable capacity. The overburden of mitochondrial complex II is speculated to account for the increased ROS production in LDH-A-inhibited cancer stem cells [83,84]. The widely studied LDH-A inhibitors include pyruvate analog, oxamate (OXM), and the NADH competitive inhibitor, gossypol. LDH-A inhibition with OXM triggers a specific tumor reduction in brain tumors by reducing ATP levels, increasing ROS production, and inducing apoptosis [85]. OXM also induces autophagy via the AKT-mTOR signaling pathway in certain cancer cell types [86]. Recent pre-clinical and clinical studies have supported the combined use of LDH inhibitors with concurrent treatments as a promising strategy in cancer therapy [85]. Combination therapy of OXM with other chemotherapeutic drugs, an including mTORC1 inhibitor and phenformin (phenethylbiguanidine; an anti-diabetic agent), have shown synergistic effects suggesting their implications in combination therapies [87,88]. A triple combination therapy of doxorubicin with metformin and OXM induced autophagy and apoptosis in colorectal cancers by downregulating hypoxia-induced HIF-1 expression [85].

Enolase (ENO-1), which converts 2-phosphoglycerate (2PG) to phosphoenol pyruvate (PEP), is emerging as a promising target for cancer therapy, partially owing to its diverse functions apart from being a major enzyme in glycolysis [89,90,91]. The overexpression of ENO-1 is associated with disease progression, metastasis-free survival, and overall survival in colorectal cancer, BC, gastric cancer, gliomas, head and neck cancer, and leukemia. ENO-1 promotes tumor cell progression via a plethora of mechanisms, including inducing angiogenesis, evading immune suppression and growth suppressors, and resisting cell death [92,93,94]. Small molecule inhibitors of ENO-1 have been shown to inhibit cancer cell growth [95,96,97]. POMHEX, a selective enolase inhibitor, has been shown to selectively inhibit the tumor cell progression of glioma cells in vivo by triggering apoptosis, showing a favorable safety profile and tolerance in non-human primates [98]. A previous study identified a potent inhibitor of ENO1, macrosphelide A, which demonstrates anti-cancer effects by simultaneously inactivating ENO1, aldolase, and fumarase [95].

6-phosphofructokinase/fructose-2,6-bisphophatase (PFKFBs) catalyzes the first irreversible step in glycolysis, which is the conversion of fructose-6-phosphate to fructose-1,6-bis-phosphate. As with most other glycolytic enzymes, PFKFB activity and expression are enhanced in many cancers. The selective inhibition of PFKFB3 displays broad anti-tumor activity in syngenic pre-clinical models and early human studies by inducing necroptotic cell death, apoptosis, cell cycle arrest, and inhibiting invasion [99,100]. A phase-1 dose escalation study for PFK-158, a first-in-human, first-in-class, small molecule inhibitor of PFKFB3, showed commendable tolerance and tumor burden reduction in pancreatic cancer, renal cell carcinoma, and adenocystic carcinoma patients [101,102,103]. The synergistic effect of PFK-158 with other FDA-approved targeted-chemotherapy agents can potentially improve their chemotherapy efficacy and is being validated. In gynecologic cancers, it was shown that PFK-158 improves lipophagy and sensitizes platinum therapy-resistant cells to carboplatin/oxaliplatin therapy [104]. Combining PFKFB3 inhibition with standard chemotherapy can thus be a novel strategy to improve the outcome in gynecologic and endometrial cancer patients who are resistant to therapy or have advanced, recurrent diseases [89].

Besides its glycolytic function, PFKFB3 is a crucial player in regulating endothelial cells, tumor angiogenesis, and tumor vascularization [105,106]. A transient inhibition of PFKFB3 in endothelial cells using 3-(3-pyridinyl)-1-(4-pyridinyl)-2-propen-1-one (3PO) induced tumor vessel normalization, impaired metastasis, and improved chemotherapy [107,108]. The PFKFB3 inhibitor, AZ67, inhibited angiogenesis in vivo, independent of glycolysis regulation [109]. A combination therapy with PFKFB3 inhibitor and VEGF inhibitor, bevacizumab, improved tumor vasculature, alleviated tumor hypoxia, normalized lactate production, and improved the efficacy and delivery of doxorubicin in glioblastoma [110].

Pyruvate kinase M2 (PKM2), which catalyzes the conversion of PEP to pyruvate, is upregulated in numerous cancers and has emerged as a critical regulator of cancer cell metabolism [111,112]. Apart from being a key enzyme in glycolysis, nuclear PKM2 regulates the expression of GLUT1 and LDH-A through positive feedback to further support glycolytic metabolism [113]. PKM2 expression is upregulated under hypoxic conditions and induces tumor angiogenesis and metastasis [114]. The association of PKM2 expression with poor prognosis and overall survival indicates that PKM2 level could serve as a prognostic or diagnostic marker for cancers [115,116,117]. PKM2 inhibition induces apoptosis and tumor regression in xenograft models of different cancer types [89] and plays a role in maintaining redox homeostasis and glutathione turnover. PKM2 inhibition also increases the efficacy of docetaxel treatment in vitro and xenograft models of LC [118,119]. In NSCLC patients who received platinum therapy in a first-line setting, tumors with low PKM2 expression showed significantly longer progression-free survival and overall survival [120]. In tumor xenograft models of NSCLC, combination therapy with PKM2 siRNA and chemotherapeutic agents increased apoptosis and inhibited tumor growth [121].

Targeting Glyceraldehyde 3-phosphate dehydrogenase (GAPDH) is being explored as an alternative approach for inhibiting glycolysis [122]. GAPDH catalyzes the first step in which energy is derived from NADH in the ‘pay-off-phase’ of glycolysis. NADH, the first molecule generated in this phase, is critical for regulating intracellular ROS and redox balance. Targeting GAPDH triggers the accumulation of glucotrioses such as glyceraldehyde-3-phosphate and dihydroxy acetone phosphate in the cells, the partial degradation of which results in the formation of cytotoxic methylglyoxal [123]. Thus, inhibiting GAPDH not only depletes ATP but also triggers cytotoxicity through the upregulation of ROS and the accumulation of methylglyoxal [122]. 3-BrPA, discussed above, is a potent inhibitor of GAPDH and was shown to deplete intracellular ATP. Additionally, 3-BrPA showed high specificity and selectivity for GAPDH both in vitro and in vivo [122,124,125].

Although targeting glycolytic enzymes can improve the efficacy of chemotherapy and radiotherapy, the ubiquitous nature of glycolysis and glycolytic enzymes presents the challenge of the systemic toxicity of glycolysis inhibition. The selective targeting of cancer-specific enzymes or enzyme isoforms and the targeted delivery of therapeutic agents could circumvent this challenge.

### 2.3. Modulating Glycolysis to Overcome Drug Resistance

Aberrant glycolysis is a major contributor to drug resistance in cancer [126,127]. The mechanism underlying glucose metabolism reprogramming-induced drug resistance is not clearly understood. Increased glucose uptake induces gemcitabine resistance in pancreatic cancer, doxorubicin resistance in BC, and cisplatin resistance in genitourinary cancers [6,128]. It is thought that glucose metabolism reprogramming in cancer cells induces DNA repair and immune suppression in the tumor microenvironment, contributing to drug resistance. Anabolic alterations could account for the increased nucleotide demand required for the efficient repair of chemotherapy/radiation-induced DNA damage. The DNA repair pathways in reprogrammed cancer cells induces the activation of several pro-tumorigenic signaling pathways, including Wnt, PI3K/AKT, NF-κB, and MAPK, triggering prolonged cancer cell survival and apoptosis resistance [89,129]. Aberrant glycolysis can also promote DNA repair by increasing nucleotide turn over by enhancing the hexosamine biosynthetic pathway (HBP) and pentose phosphate pathway (PPP) [130,131]. By limiting pyruvate flux into OXPHOS, upregulated glycolysis also enables cancer cells to reduce the ROS accumulation in cells, another mechanism by which metabolic reprogramming contributes to resistance to therapy. Increasing evidence also suggests that the activation of Wnt, PI3K/AKT, and Notch pathways activate autophagy which also contributes to cancer cell survival and resistance to therapy, whereas inhibiting autophagy sensitizes cancer cells to therapy [89,132,133]. Autophagy, thus, downregulates cell metabolism leading to cancer cell quiescence and survival, inducing radio-resistance [134].

Metabolic changes in the tumor cells happen hand-in-hand with similar reprogramming of the tumor microenvironment. This metabolic reprogramming induces immunosuppression and immune escape of cancer cells and contributes to the development of resistance to chemotherapy and radiotherapy [135,136]. The upregulation of glycolytic enzyme HK2 suppresses the mTOR-S6K signaling pathway and blocks chemotherapy-induced apoptosis by binding to voltage-dependent anion channels, and suppresses the formation of mitochondrial permeability transition pores, contributing to chemoresistance. Aberrant glycolytic pathways in cancer stem cells also play critical roles in contributing to resistance to therapy via enhancing cancer cell stemness by activating the PI3K/AKT pathway and upregulating the stem cell-like properties [89]. Enhanced exosomal secretion from cancer stem cells also activates neighboring cancer cells toward stemness and promotes chemo/radio-resistance [137,138]. The overexpression of ENO-1 in cancer cells can also contribute to cisplatin resistance in different cancer types [139] and is considered a biomarker to predict prognosis and drug resistance in cancers [85,140].

Glycolytic inhibitors are reported to sensitize cancer cells to chemotherapy and radiotherapy, thereby overcoming resistance to therapy. 3BrPA aids in dissociating HK2 from the mitochondrial complex and improves therapy response to daunorubicin [15]. A combination therapy of curcumin and docetaxel has demonstrated improved drug response and tolerance in a clinical study in prostate cancer patients [141]. 2-DG, a glycolytic inhibitor that modulates several glycolytic enzymes, restores sensitivity to adriamycin in ER+ BC cells. In HER2+ BC, trastuzumab inhibits tumor growth by downregulating heat shock factor 1 (HSF1) and LDH-A, thereby inhibiting glycolysis. A combination therapy of trastuzumab with LDH-A siRNA-mediated glycolysis inhibition synergistically inhibited tumor growth in trastuzumab-resistant breast cancer cells, suggesting their usefulness in overcoming drug resistance. Further, the combination therapy of trastuzumab with glycolytic inhibitor 2-DG and oxamate increased the sensitivity of ErbB2-positive cancer cells to therapy. An allosteric inhibitor of phosphoglycerate mutase (PGAM), a glycolytic enzyme that converts 3-phosphoglycerate to 2-phosphoglycerate, has been shown to overcome erlotinib resistance in NSCLC. PGAM inhibition alters the ERK and AKT signaling pathways and induces oxidative stress and ROS production to overcome erlotinib resistance [142].

ENO-1 overexpression has been associated with chemoresistance in prostate and pancreatic cancer cells [102,143,144,145]. Cisplatin-resistant gastric cancer cells also exhibit enhanced glycolysis by upregulating ENO-1. ENO1 inhibition in cisplatin-resistant cells increased sensitivity to the therapy by activating apoptotic pathways or inducing autophagy [96]. In ovarian cancer cells, inhibiting ENO-1 expression increased cell senescence and improved cisplatin resistance [146]. Hypoxia-induced resistance to gemcitabine is a critical issue in PDAC treatment. A recent study demonstrated that the shRNA-based downregulation of ENO-1 modulated redox homeostasis, increased intracellular ROS concentration, and sensitized resistant PDAC cells to gemcitabine treatment. In ovarian cancer models, PFKFB3 inhibitors, 3-PO and PFK-158, impaired metabolic reprogramming-induced stemness and chemoresistance, possibly by modulating apoptosis via the NF-κB pathway [147,148].

PKM2 can contribute to chemoresistance against cisplatin and gemcitabine treatment in different cancer types. PKM2 overexpression has been reported to be a biomarker for cancer resistance. PKM2 regulates the DNA repair mechanism in addition to glucose metabolism and induces resistance to genotoxic damage, driving treatment resistance [149,150]. Targeting PKM2 sensitizes cancer cells to treatment. In NSCLC, shRNA-based silencing of PKM2 enhanced radiation-induced autophagy in vitro and in vivo [149] and increased the sensitivity to docetaxel treatment [119]. PKM2 expression also correlated with a resistance to platinum-based therapy in colorectal cancer [151].

A few studies, however, have reported contradicting results, where PKM2 activation was shown to act as a chemosensitizer in some cancer types. In a study by Anastasiou et al., an increase in intracellular ROS concentration in response to therapy was shown to inhibit PKM2, which in turn diverted the glucose flux into PPP generating redox potential for the detoxification of ROS. These regulatory properties of PKM2 confer an additional advantage to cancer cells to tolerate therapy-induced oxidative stress. The endogenous expression of oxidation-resistant-PKM2 mutants increased oxidative stress and impaired tumor progression [152]. PKM2 activation could thus be an attractive strategy in cancer therapy. High levels of PKM2 activate the mTOR-HIF1α pathway and are associated with a positive chemotherapy response in cervical cancer patients treated with cisplatin-neoadjuvant chemotherapy [153,154]. The high expression of PKM2 has been shown to enhance drug response to epirubicin and 5-fluorouracil in BC [155], whereas a decrease in PKM2 levels/activity contributes to cisplatin/oxaliplatin resistance in cervical cancer, colorectal cancer, and gastric cancer cells [153,156,157].

The development of resistance is frequently encountered in cancer treatment, and the link between cancer cell metabolism and the development of resistance is becoming more apparent. Targeting metabolic enzymes is an efficient strategy to re-sensitize the resistant cells to chemotherapy and radiotherapy. However, the clinical application of glycolysis inhibition to overcome drug resistance has remained limited. Future studies should identify the key metabolic shifts that contribute to the development of drug resistance and explore their potential as drug targets to improve the sensitivity of cancers to chemotherapeutic agents and radiotherapy and to overcome the development of resistance. Figure 1 summarizes how targeting glycolysis can be used to modulate cancer therapy (Figure 1).

## 3. Targeting Glycolysis to Enhance Immunotherapy

The advances in our understanding of the remarkable potential of the immune system to fight cancer have garnered tremendous attention on immunotherapy for cancer. Arguably, immunotherapy is now being considered one of the most promising therapeutic strategies for several types of cancers, and immune checkpoint blockade (ICB)- and adoptive-cell therapy (ACT)-based therapeutic strategies have been approved for several cancers [158,159,160]. However, several reports have shown that a high percentage of patients initially fail to respond to these interventions or acquire resistance in the long run [161,162], limiting the application of these promising strategies. Several studies have reported that the metabolic reprogramming of the cancer cell that leads to the development of an immunosuppressive tumor microenvironment is one of the main contributors to the reduced efficacy of immunotherapy [163]. Further, it has been suggested that metabolic interventions can significantly enhance the efficacy of immunotherapy [164,165]. Therefore, understanding the challenges the immune cells face in the harsh immunosuppressive tumor microenvironment and identifying strategic interventions to overcome these challenges would contribute to improving immunotherapy.

### 3.1. Glucose Metabolism in the Immune Cells of the Tumor Microenvironment

It is known that cancer cells undergo a metabolic reprogramming called the ‘Warburg effect’ or aerobic glycolysis in response to hypoxia and oncogenic signals, such as Myc and PI3K [165]. This preference for glycolysis is also shared by other rapidly proliferating cells in the TME, including effector T-cells and M1-like macrophages, to satisfy their increased energy requirements [166]. In contrast, other cells in the TME, including memory T-cells, regulatory T-cells (Tregs), and M2-like macrophages, rely on fatty acid oxidation (FAO) to satisfy their energy needs.

The naïve T-cells utilize TCA-coupled OXPHOS as their primary energy source [167]. On MHC activation, the T-cell receptor (TCR) and CD28 activate the PI3K-AKT-mTORC1 and Myc signaling pathways and induce metabolic reprogramming [167]. The effector T-cells upregulate aerobic glycolysis and enhance their anabolic metabolism for cancer-killing and clone expansion. In addition, the glycolytic intermediates support effector T-cell activation and cytokine generation. Phosphoenolpyruvate (PEP), a glycolytic metabolite, blocks sarco/endoplasmic reticulum Ca^2+^-ATPase (SERCA)-mediated endoplasmic reticulum calcium uptake and the nuclear factor of activated T-cells (NFAT) signaling, enabling TCR signaling [168]. In addition, phosphoenolpyruvate carboxykinase 1 (PCK1) catalyzes the conversion of oxaloacetate (OAA) into PEP, and the overexpression of PCK1 enhances the cancer-killing functions of adoptive transferred CD4+ and CD8+T-cells [168]. However, although CD8+ T-cells experiencing continuous stimulation or hypoxia differentiated into functional effectors in vitro, it rapidly drove T-cell dysfunction and exhaustion [169].

Similar to the CD8+ T-cells, the functions of the CD4+ T-cells are also affected by specific metabolic reprogramming. Increased glycolysis promotes IL-2, TNFα, and IFN secretion in the CD4+ T-cells, and the inhibition of glycolysis drives the functional and metabolic exhaustion of the CD4+ T-cells [170,171]. In line with this, the inflammatory CD4+ T-cells (Th1 and Th17) show enhanced glycolysis. The Th17 cells exclusively express the pyruvate dehydrogenase (PDH) inhibitor, pyruvate dehydrogenase kinase isozyme 1 (PDHK1), which, when downregulated, leads to the selective reduction of Th17 cells [172]. On the other hand, Tregs show upregulated OXPHOS and FAO [172], and their expression of FOXP3 inhibits Myc and attenuates PI3K-AKT-mTORC1 axis-mediated activation of glycolysis and increases oxidation and catabolic metabolism, rendering a survival advantage in the TME [64,173]. Furthermore, the Tregs utilize lactate metabolism, and culturing them in high-glucose conditions decreases their stability [167]. The mechanisms by which the low levels of oxygen, high levels of lactate, and the high competition for glucose potentially contribute to T-cell dysfunction in the tumor microenvironment is been summarized in Figure 2.

The M1-like macrophages preferentially utilize glycolysis to sustain the inflammatory phenotype [174], while the M2-like macrophages depend on TCA and FAO to maintain an immunosuppressive phenotype [175]. The highly acidic environment of melanoma was found to induce tumor-associated macrophages (TAM) toward a cancer-promoting phenotype [176]. The tumor-associated neutrophils (TAN), on the other hand, exhibit pro- or anti-tumor effects in different cancer microenvironments. In pancreatic ductal adenocarcinoma (PDAC), it was shown that the TANs undergo an LDH-A-mediated glycolytic switch and exhibit a tumor-promoting phenotype [177]. Triple-negative BC (TNBC) cells with an accelerated glycolysis support myeloid-derived suppressor cell (MDSC) development and facilitate CD8+T-cell inhibition and cancer progression [178]. In addition, an increase in lactate generation enhances the tumor-promoting capacity of MDSCs [179].

The TME is characterized by a decrease in nutrients, insufficient vasculature, increased lactate accumulation, and hypoxia, conditions that affect the cancer-killing capacity of the T-cell. The cancer cells and the immune cells compete for glucose utilization, and the activated glycolysis of the cancer cells endows them with an advantage over the immune cells, impairing the function and survival of the effector T-cells. It has been shown that the expression of glycolysis-related genes, such as *ALDOA*, *ALDOC*, *ENO2*, *GAPDH*, *GPI*, and *PFKM*, negatively correlates with T-cell infiltration in melanoma and NSCLC [180]. The lower availability of glucose in the TME affects the glycolytic capacity of the T-cells [168], inducing T-cell exhaustion [181]. Furthermore, the lack of glucose also affects mitochondrial functions, promoting terminal CD8+ T-cell exhaustion [182]. In addition to the competition for glucose, the increased production of lactate due to the increased rate of glycolysis contributes to the immunosuppressive character of the TME. Lactate metabolism has been shown to contribute to oncogenesis [183], and LDH has been reported to be a marker for poor prognosis and immunotherapy efficacy [184,185]. The low pH in the TME triggers reduced CD25 and TCR expression in the effector CD8+ T-cells and inactivated STAT5 and ERK signaling, affecting anti-tumor immunity [186]. Lactate also affects TCR signaling [187], and exposure to large amounts of lactate makes the Tregs switch to OXPHOS for regenerating NAD+; however, the Tregs fail to maintain the NAD+/NADH balance through this pathway [64]. Moreover, lactate in the TME promotes the expression of pro-inflammatory cytokines, including IL-23 and IL-17, supporting tumorigenesis and impairing anti-tumor immunity [188]. The poor vasculature and the increased metabolism of the cancer cells contribute to the formation of a hypoxic environment, further affecting anti-tumor immunity. Hypoxia triggers epigenetic reprogramming of effector T-cells, reducing their functional capabilities [189,190]. Although hypoxia-inducible factor-1α (HIF-1α), expressed in response to hypoxia, can induce Tregs and bind to the promoter region of the FOXP3 to promote transcription [191] (22988108), it contributes to the development of an immunosuppressive TME by enhancing cancer-promoting immune cell functions [192,193].

### 3.2. Signaling Mechanisms Regulating Glycolysis

The signaling pathways that modulate glycolysis in the immune cells can be potentially targeted to improve their anti-tumor functions, and drugs such as metformin and phenformin have been extensively tested in the clinical setting [194,195,196]. The liver kinase B1 (LKB1)-AMPK pathway and the PI3K-AKT-mTOR axis are the two primary signaling mechanisms that modulate glycolysis in the cell. The LKB1-dependent kinases regulate metabolic pathways by targeting several effectors, including AMPK [197]. LKB1 loss upregulates GLUT1 and hexokinase 2 (HK2), increasing T-cell glycolytic transcription and flux [198].

On the other hand, the deregulation of the PI3K-AKT-mTOR axis promotes HIF-1 activation and GLUT1 expression [199,200]. PI3K activation increases AKT phosphorylation and glycolytic flux via LDH-A in T-cells [201]. AKT is the primary glycolysis regulator in both cancer and immune cells. AKT activation induces GLUT1 and LDH-A expression [202], activates HK1 by promoting HK2 and PFK2 phosphorylation [203], and inhibits PDH by activating pyruvate dehydrogenase kinase-1 (PDK1) [204], resulting in activated glycolysis. mTOR regulates the expression of HIF-1α, the major transcription factor regulating several glycolytic enzymes and GLUT1 [199]. mTOR kinases determine effector and memory CD8+ T-cell fates, and blocking mTOR promotes T-cell effector functions [205,206].

### 3.3. Targeting Glycolysis to Improve Immunotherapy

Adoptive-cell transfer (ACT) and immune checkpoint blockade (ICB) are the two primary strategies for immunotherapy. The interplay between anti-tumor immunity and cancer metabolism suggests that combining immunotherapy with glycolysis-targeted therapy is a promising strategy to improve treatment efficacy. ACT utilizes therapeutic-modified immune cells to directly boost anti-tumor immunity. Ex vivo-expanded T-cells and T-cells engineered to express antigen-specific TCRs or chimeric antigen receptors (CARs) are primarily used for ACT [207]. ACT has shown promising efficacies, particularly with CD19-specific CAR-T cells, in the treatment of B-cell acute lymphoblastic leukemia and B-cell lymphomas [208]. However, responses in other cancers have been poor. It has been suggested that optimizing T-cell metabolism to support robust initial and durable T-cell responses for target cells and the TME would improve and broaden their applicability.

CAR-T cells can be engineered to express specific signaling domains, and endowing them with specific properties can tailor them for effector activity and long-lasting memory. The two FDA-approved CARs carry a CD19-targeting extracellular domain coupled with an intracellular signaling domain from CD3ζ and the co-stimulatory molecules CD28 or 4-1BB [209]. Linking TCR signaling with the co-stimulatory signals enables the CARs to elicit effector functions in the absence of additional inflammatory or co-stimulatory stimuli [210]. CAR-T cells carrying the CD28 domain have increased glycolysis and show enhanced effector responses but are short-lived. On the other hand, CAR-T cells expressing the 4-1BB domain show elevated OXPHOS and FAO and display a memory phenotype [211]. This is in line with the normal physiological functions of the co-stimulatory molecules. CD28 stimulation activates PI3K/AKT/mTOR, promoting glycolysis and effector differentiation, whereas 4-1BB activates AMP-driven FAO and OXPHOS [211]. Based on this understanding, it has been proposed that introducing signaling mutations could improve CAR-T cell effector functions and survival. For example, it was shown that mutations of the YMNM signaling motif of CD28 increase CAR-T cell survival and reduce T-cell exhaustion, enabling enhanced tumor control [212]. Furthermore, additional co-stimulatory molecules modulating T-cell metabolism may be incorporated into CAR constructs to improve T-cell function. For example, ICOS, a CD28 family member, promotes glycolysis and mTORC1 activity in T follicular helper cells [213]; GITR agonists enhance cellular metabolism to support CD8+ T-cell proliferation and effector cytokine production [214]; OX40 is associated with enrichment of glycolysis and lipid metabolism transcripts; and OX40 agonists enhanced lipid uptake in Tregs [215]. Similar to the CAR-T cells, T-cells procured from tumors and expanded ex vivo, or modified to express engineered TCRs, can also be optimized through metabolic manipulations, and T-cells with low glycolysis rates can be generated or selected for longevity while retaining effector functions [216].

The in vitro stimulation and the T-cell engineering phases of the ACT strategy provides the added advantage of the opportunity to modify T-cell metabolism and mitochondria without affecting other cells and tissues and thus prevent any potential boosting of the cancer cell metabolism from the intervention. It was also shown that blocking glutamine metabolism increases T-cell function [217]. Furthermore, supplementing the culture medium with glutamine antagonist 6-Diazo-5-oxo-l-norleucine (DON) enhanced CAR-T cell FAO and reduced glycolysis, making the CAR-T cells remain in a more undifferentiated state [218]. Inhibiting glycolysis using the HK-2 inhibitor, 2-DG, before ACT induced a memory-like phenotype in the T-cells, enabling a more efficient control of tumors and prolonged survival in an animal model [219]. Similarly, CD19-CAR-T cells treated with AKT inhibitors showed reduced glycolysis and a memory-like phenotype and had robust tumor elimination potential [220]. Conversely, modulating mitochondrial OXPHOS and FAO would enable CAR-T cell longevity and the continued expression of a memory-like phenotype. Furthermore, a transient glucose restriction followed by glucose re-exposure could enhance the tumor-clearing efficacy of CD8+ T-cells [221].

Unlike ACT, which directs anti-tumor immunity through pharmaceutical interventions, ICB aims to modify inhibitory signals to activate endogenous anti-tumor-specific T-cells. Furthermore, ICB is primarily targeted against solid tumors, which show a greater influence of the TME in modulating T-cell metabolism. In addition to the initial challenge of T-cells infiltrating the tumors, ICB must overcome several obstacles, including tumor-infiltrating lymphocyte (TIL) exhaustion, the upregulation of inhibitory receptors and epigenetic modifications, metabolic adaptations resulting in nutrient deficits, impaired translocation of GLUT1 to the cell surface, the downregulation of glycolytic enzymes, GAPDH and ENO-1, and dysregulated and fragmented mitochondria with increased ROS generation [222,223,224,225]. It was reported that the TIL inflammatory function could be enhanced by rescuing TIL metabolism by expressing PCK to promote gluconeogenesis and replacing the intracellular glycolytic intermediates or by improving mitochondrial metabolism by treating with pyruvate or acetate [225]. The immunosuppressive environment of the TME and chronic antigen stress direct the T-cells to an exhausted state, characterized by the expression of immune checkpoints and decreased cytotoxicity. The checkpoint molecules, including CD28, CD40L, and cytotoxic T lymphocyte-associated protein-4 (CTLA-4), along with TCR signaling, drive TIL exhaustion and contribute to the persistence of the exhausted state. The concurrent metabolic adaptations further enhance the immunosuppressive effects of the checkpoints. Thus, targeting glycolytic regulators and metabolites that support the immune checkpoint-directed T-cell inhibition is a potential strategy to improve the efficacy of ICB.

The most well-known and commonly targeted immune checkpoints in cancer are the programmed cell death protein-1 (PD-1) and CTLA-4. CTLA-4 is expressed primarily on T-cells and plays an immunosuppressive role during the initial phase of T-cell activation and downregulates T-cell activation-triggered glycolysis. PD-1 is activated after TCR activation and impedes glucose uptake and glycolysis while promoting FAO. Thus, these checkpoint signals prevent T-cell activation and inflammation. It was shown that PD-1–deficient T-cells maintain higher metabolic activity in chronic infection [226,227]. Thus, blocking PD-1 and CTLA-4 relieves PI3K/AKT/mTORC1 signaling and allows increased T-cell stimulation and metabolism, inducing an effector-like phenotype. In addition, this metabolic shift induces epigenetic reprogramming inducing effector functions and longevity. Although PD-1 and CTLA-4 are the most extensively targeted ICB candidates, several other co-inhibitory and co-stimulatory molecules modulate T-cell metabolism. Notably, T-cell immunoglobulin mucin receptor 3 (TIM-3) downregulates glycolysis and GLUT1 expression [228], while LAG3 downregulates OXPHOS [229]. Similarly, inhibiting 4-1BB and OX40 enhances T-cell OXPHOS and promotes effector function and longevity [230]. Taken together, immune checkpoint molecules induce metabolic dysfunction and thus affect anti-tumor immunity, suggesting that combining ICB with metabolic regulators is a potential strategy to improve treatment efficacy.

### 3.4. Glycolysis-Targeting Therapies to Improve Immunotherapy Efficacy

mTOR is an oncogenic molecule that contributes to the regulation of metabolism in TILs. The inhibition of the mTOR signaling axis downregulates the malignant phenotype of cancer cells. Owing to their inhibitory effects, several rapamycin analogs have been approved for treating cancers [231,232]. However, it was shown that mTOR inhibition could diminish anti-tumor immunity [194] as these inhibitors directly affect the lineage differentiation-determining glycolytic activity in T-cells. It was shown that rapamycin suppresses Th17 differentiation and promotes Treg differentiation under TGFβ induction [233,234]. Further, the activation of AKT-mTORC1 signaling was associated with T-cell function restoration and the reduced expression of PD-1 and TIM-3 [235]. Additionally, the over-activation of mTORC1 affects the immunosuppressive function of Tregs, while low mTORC1 levels enhance Treg activity [236]. These suggest that an optimized inhibition of the PI3K-mTORC1 signaling axis is critical for improving the efficacy of immunotherapies.

Metformin has shown promising effects in different cancers [195] and has been shown to regulate metabolism by interacting with AMPK, the PI3K-AKT-mTOR axis, and HIF-1α [237,238]. Additionally, metformin promotes the cancer-killing capacity of CD8+ T-cells by modulating glycolysis [239,240,241] and downregulates immune checkpoint expression and glycolytic flux through HIF-1α inhibition [242,243]. Furthermore, it was shown that metformin stops the cancer cells from using the lactate and ketone bodies produced by cancer-associated fibroblasts as nutrients and thus suppresses cancer progression [244]. A recent study combined 2-DG, an HK inhibitor [245], BAY-876, a GLUT-1 inhibitor, and chloroquine and developed the nano-drug, D/B/CQ@ZIF-8@CS, which inhibited glycolysis and improved anti-CTLA-4 immunotherapy by reducing Treg metabolic fitness [246]. Tregs pretreated with 2-DG showed enhanced inhibition of T-cell proliferation in ovarian cancer [247]. Furthermore, HK upregulates PD-L1 expression in cancer cells, and combining the HK inhibitor, Lonidamine, with anti-PD-1 therapy improved cancer cell elimination in a mouse model [248].

Knocking out glucose-6-phosphate isomerase (GPI), the enzyme that catalyzes the conversion of glucose-6-phosphate (G6P) to fructose-6-phosphate (F6P), upregulated OXPHOS and sustained the survival of cancer cells [245]. Additionally, GPI inhibition selectively eliminated inflammatory encephalitogenic and colitogenic Th17 cells without affecting the homeostatic microbiota-specific Th17 cells [249]. However, it remains unknown whether GPI-targeted therapies would improve the efficacy of immunotherapy.

The GAPDH inhibitor, dimethyl fumarate (DMF), promotes the oxidative PPP and inhibits glycolysis and OXPHOS in cancer cells. This reduces the competition between cancer cells and T-cells for glucose consumption and promotes the efficacy of ICB and IL-2 therapy [250]. Low-dose osimertinib was shown to inhibit GAPDH and tumor endothelial glycolysis and promote vascularization and immune cell infiltration and thus improve the efficacy of anti-PD-1 therapy [251]. Inhibiting fructose-2,6-bisphosphatase 3 (PFKFB3), which is upregulated in several cancers, repressed glycolysis and upregulated PD-L1 expression [193]. On the contrary, glucose deficiency upregulated PD-L1 through the EGFR/ERK/c-Jun pathway, leading to the upregulation of PFKFB3, and promoted glycolysis [252,253]. This suggests the existence of a positive feedback loop between metabolism and checkpoint molecules [254]. A dual-target drug comprising paclitaxel and the PFKFB3 inhibitor, PFK15, blocked cancer-associated fibroblast-mediated cancer cell growth and reduced the lactate concentration in the TME [193]. PFK15 was also shown to upregulate PD-1 and LAG-3 expression in the context of type 1 diabetes [171]. Pyruvate kinase isoform M2 (PKM2), the final rate-limiting enzyme in glycolysis, promotes PD-L1 expression in macrophages, DCs, and tumor cells and contributes toward accelerated tumor progression [255]. High PKM2 expression was associated with a poor prognosis of pancreatic ductal adenocarcinoma, and the knocking down of PKM2 improved the efficacy of anti-PD-1/PD-L1 therapy [256].

ENO-1, which catalyzes the conversion of 2-phosphoglycerate to PEP and also acts as a plasminogen receptor and a DNA-binding protein, was shown to be overexpressed in several cancers. [64]. A pan-cancer analysis showed that ENO-1 expression correlated with immune cell infiltration, including B cells, CD8+ and CD4+ T-cells, macrophages, neutrophils, and dendritic cells [257]. The presence of autoantibodies against ENO-1 correlated with a better prognosis in PDA, suggesting that ENO-1 was a good molecular candidate for improving immune cell response to cancers [258]. Antibodies against ENO-1 were detected in approximately 60% of patients with PDAC, and ENO-1-specific T-cell responses are observed in patients who have the anti-ENO-1 antibodies. In line with this, an ENO-1 DNA vaccine induced an antibody and cellular response and increased the median survival in mouse models of PDA [259]. ENO-1-targeting DNA vaccines have shown prophylactic and therapeutic potential in PDAC mouse models by inducing complement-dependent cytotoxicity and immune cell response [144,259,260]. In a spontaneous mouse model of PDAC, co-treatment with gemcitabine and ENO-1 DNA vaccine enhanced CD4 anti-tumor activity and impaired tumor progression [261,262]. Additionally, a recent study showed that targeting ENO-1 using specific antibodies targets multiple TME niches involved in prostate cancer (PC) progression and bone metastasis via a plasmin-related mechanism [263]. These show the potential of ENO-1 targeted therapies in improving the efficacy of immunotherapies.

Optimizing T-cell metabolism is a promising strategy for improving cancer immunotherapy. Metabolic modification can potentially increase stemness and long-term memory, enhance effector functions, and reduce T-cell exhaustion. Future studies should identify key metabolic transitions and regulatory steps that are differently regulated in cancer cells and immune cells and develop effective targeting strategies to enhance the synergistic effects of metabolic modulation and cancer immunotherapy [264].

## 4. Targeting Glycolysis to Enhance Hormonal Therapy

Hormonal therapy has shown remarkable advancement as a therapeutic strategy for cancers dependent on hormones, especially in breast, prostate, and other gynecological cancers. Aromatase inhibitors (AI), estrogen receptor (ER) antagonists, ER modulators, anti-estrogens, and GnRH agonists are effective therapeutic drugs and have shown high success rates in patients with hormone-sensitive recurring or metastatic gynecologic malignancies [265]. Hormone therapy interferes with the hormone-dependency of cancer cells by limiting hormone production in the body [266]. While hormonal therapy has improved survival and reduced recurrence in different cancer types [266], de novo or acquired resistance to hormonal therapy is a major clinical problem that requires the development of innovative strategies [264]. Resistance to hormonal therapy invariably occurs in most patients with ER+ metastatic BC and castration-resistance PC (CRPC) [267]. Metabolic reprogramming is an inherent feature of endocrine-resistant cancer cells, implicating that combination therapy with metabolic regulators and conventional hormonal therapy might be beneficial in overcoming resistance [268]. However, it is unclear whether metabolic rewiring is a cause or consequence of endocrine resistance, and several studies are investigating the cross-talk between hormone signaling and cancer cell metabolism [269]. Somatic mutation in estrogen receptors is related to the clinical development of the resistance to hormone therapies [268,270,271,272]. The Y537S mutation in ER-α enhanced mitochondrial metabolism and glycolysis in BC cells. The Y537S mutation is also associated with poor clinical outcomes, suggesting that enhanced glucose metabolism is a highly conserved mechanism of endocrine resistance [268].

Elevated glucose levels resembling hyperglycemia in BC cells have been attributed to a reduced response to tamoxifen therapy and could act as a marker for responses to hormonal therapy [273,274]. An increased glycolytic rate is a characteristic feature of tamoxifen-resistant cells, and inhibiting glycolysis is expected to restore tamoxifen sensitivity [273,275]. Elevated glycolysis in BC cells is also associated with mitochondrial malfunction and upregulated AKT/mTOR and HIF-1α signaling pathways. Tamoxifen-resistant BC cells escape cell death by increasing autophagy through the inactivation of TOR-S6K via the HK2 pathway [276]. Glycolytic inhibition by the knockdown of HK2 or 3BrPA treatment downregulated AKT/mTOR signaling and could be a therapeutic strategy to overcome tamoxifen resistance in BC [277]. A recent study investigated the potency of a combination therapy employing low-dose tamoxifen (ERα antagonist) and metabolism inhibitors, 2-DG and CB-839 (glutaminolysis inhibitor), in improving the anti-proliferation effect in tamoxifen-resistant ERα-positive BC cells. The triple combination showed superior cell growth inhibition by inducing apoptosis and c-Myc downregulation; however, a combination of tamoxifen with 2-DG did not show significantly strong inhibition of cell viability [278,279]. The pharmacological inhibition of glycolysis with PFK-158, a PFKFB3 inhibitor, with tamoxifen or fulvestrant has been explored as a potential therapeutic intervention to overcome endocrine resistance. PFKFB3 upregulation, with an elevated basal expression of PFKFB3 mRNA, is observed in endocrine-therapy-resistant BC cells and is associated with adverse recurrence-free survival in BC patients. The anti-tumor effect of PFK-158 is exacerbated when combined with tamoxifen and fulvestrant treatment [280]. PFKFB3 inhibition activated necroptotic markers receptor-interacting kinase 1 (RIPK1) and mixed lineage kinase domain-like pseudokinase (MLKL), implicating the possible mechanism of PFK-158-induced cell death [281]. In a long-term estrogen deprivation model (LTED) of AI resistance, cancer cells were demonstrated to have increased glycolysis dependency. The inhibition of glycolysis with HK2 inhibitors, along with AI, and letrozole, reduced cell viability [282]. Dietary interventions that target metabolic rewiring have also been shown to improve the efficacy of endocrine therapy in liver metastatic BC patients. Metastatic burden in the liver increases with increasing carbohydrate percentage in the diet. A fasting-mimicking diet increased the efficacy of fulvestrant treatment and reduced the metastatic burden in BC liver metastatic models, providing a proof-of-concept for a more straightforward strategy to circumvent drug resistance, with potential applicability in other cancer types as well [283].

Metabolic reprogramming is emerging as a crucial mechanism contributing to resistance to endocrine therapy in PC [284]. The expression of key glycolytic enzymes, including LDH-A, MCT-4, and GLUT1, is elevated in mCRPC patients [285]. Glycolysis inhibition by targeting GLUT1 plays an important role in drug response in prostate PC [286]. In PC, elevated androgen levels increase glucose uptake and upregulate the expression of GLUTs, implying a cross-talk between androgen signaling and glycolytic pathway, a mechanism that protects PC cells from glucose-deprivation-induced oxidative stress [287,288]. NF-κB-mediated GLUT1 overexpression and upregulated glucose metabolism are associated with enzalutamide resistance in PC [289,290]. In xenograft models of CRPC and enzalutamide-resistant PC patients, GLUT1 inhibition significantly reduced tumor volume and displayed synergistic effects with androgen receptor (AR)-targeted therapy [286]. Glycolytic inhibitors, gossypol (LDH-A inhibitor), and AZD3965 (MCT-1 inhibitor) are currently in clinical trials as potential glycolysis-targeting agents in mCRPC. Progesterone treatment was reported to have an anti-tumor effect in glioblastoma multiform (GBM) in vitro and in vivo and improved the efficacy of temozolomide [291]. Recently it was shown that high-dose progesterone treatment inhibits GBM growth by inhibiting the key modulators of glycolytic metabolism. This early observation highlighted the potential of progesterone in metabolic reprogramming; however, more direct evidence is essential to validate this, and future studies should determine the synergistic effect of direct glycolytic inhibitors and progesterone in GBM treatment [291].

Endocrine resistance remains a major clinical barrier that requires the development of novel strategies to circumvent the resistance. Several mechanisms that contribute to endocrine resistance have been identified. Metabolic rewiring is frequently observed in most cancer cells that exhibit resistance, and targeting glucose metabolism with well-established glycolytic inhibitors has shown to enhance the sensitivity to endocrine therapy in breast and PC models. The mutual interplay between glucose metabolism and androgen receptor/ER signaling implies that combination approaches of endocrine therapy with metabolic modulators could be a standard-of-care to overcome resistance. Dietary interventions that modulate glucose metabolism have also been demonstrated to be an interesting strategy for evading resistance to therapy. Well-designed clinical trials are urgently needed to elucidate the clinical utility of the strategies mentioned above and to develop metabolic drugs as routine standard-of-care in endocrine-resistant cancer patients in clinical settings.

## 5. Targeting Glycolysis to Improve Photodynamic Therapy

Photodynamic therapy (PDT) is a relatively new, minimally invasive therapeutic procedure that relies on the selective accumulation of a photosensitive compound in the cancer cells, which, on excitation with light of an appropriate wavelength, would generate ROS, predominantly singlet oxygen, within the cancer cells, and eventually kill the cancer cell, with minimal damage to the surrounding tissue [292,293,294]. Although PDT is widely used to treat several cancers, its efficacy is limited by several factors, including the effective irradiation of deep tissue. Therefore, several studies have attempted to improve the efficacy of PDT by combining it with other chemotherapeutic agents. It has been shown that glycolytic inhibitors disrupt cancer cell metabolism, elevate the cellular ROS level, and disrupt the mitochondria, resulting in cell death [295]. Therefore, when combined with PDT, glycolytic inhibitors could, in theory, enhance the levels of cellular ROS and thus trigger increased cancer cell death.

5-aminolevulinic acid (5-ALA) is one of the most commonly used photosensitizers for photodynamic therapy. 5-ALA is a naturally occurring non-proteinogenic δ-amino acid synthesized in the mitochondria by the condensation of glycine and succinyl-CoA by mitochondrial 5-ALA synthase (ALAS). This is the first committed step toward heme biosynthesis. The final precursor of heme is Protoporphyrin IX (PpIX), which is a highly potent photosensitizer. The exogenous supplementation of 5-ALA overrides the normal feedback inhibition of ALAS and results in the accumulation of PpIX selectively in cancer cells, owing to the differences in the heme biosynthesis pathway enzyme activities between the cancer cells and normal cells. This cancer-specific accumulation of PpIX is exploited for selectively purging cancer cells by PDT and for the visualization of tumor tissue by photodynamic diagnosis (PD). 5-ALA was approved by the U.S FDA in 2017 as an adjunct for the visualization of malignant tissue in grade III and IV glioma (NDA 208630/SN0014) and is currently used in the clinic to guide the resection of malignant glioma and glioblastoma. In addition to its role as a precursor of PpIX, 5-ALA has been reported to enhance aerobic bioenergetics [296], promote mitochondrial protein expression, and stimulate heme-oxygenase-1, triggering heme degradation [297]. Figure 3 describes the heme biosynthetic pathway, and how it is correlated with glycolysis.

A recent study showed that 5-ALA is a potent competitive inhibitor of LDH with efficacies comparable to oxamate (OXM) and tartronate (TART) [298]. They showed that treatment with 5-ALA induced glycolysis inhibition and triggered cell death in glioblastoma cell lines. Further, it was shown that up to 25% of the 5-ALA was used for glycolysis inhibition in these cells, leaving a lower amount of 5-ALA for conversion to PpIX and subsequent use as a photosensitizer for PDT and PD. Treating the glioblastoma cells with an LDH inhibitor before 5-ALA treatment enhanced the efficacy of PDT by 15%. Precise delineation of the tumor–normal tissue is of critical importance, especially in brain cancer surgeries, and a 15% increase in PD efficacy is a significant improvement.

Another study showed that exogenous 5-ALA suppressed oxidative metabolism and glycolysis and reduced cell proliferation in ovarian and BC cells [299]. Further, 5-ALA also destabilized Bach1 and inhibited cancer cell migration. The study also showed that 5-ALA-induced suppression of oxidative metabolism and glycolysis was mediated through different mechanisms in BC and ovarian cancer but involved Bach1 destabilization, AMPK activation, and the induction of oxidative stress. Additionally, an inverse relationship between oxidative metabolism and ALA sensitivity was revealed.

It was also shown that the administration of 5-ALA for 6 weeks reduced the plasma glucose levels in rats without affecting their plasm insulin levels and induced HO-1 expression in the liver and white adipose tissue [297]. An increase in HO-1 indicates an increase in heme in the liver, which promotes the formation of nuclear receptor subfamily 1 (Rev-Erbα) with its corepressor nuclear receptor co-repressor 1 (NCOR), which in turn downregulates the enzymes involved in gluconeogenesis such as PEPCK and G6Pase, resulting in reduced glucose production in the liver [300]. On the other hand, 5-ALA administration enhances glucose metabolism in the adipocytes by decreasing the total amount of adipose tissue or decreasing the number of mitochondria in the adipose tissue [301]. It has been shown that the induction of peroxisome proliferator-activated receptor γ coactivator 1-α (PGC-1α), a master transcription coactivator, increases heme synthesis by upregulating ALAS [302]. Glucose intake repressed PGC-1α mediated ALAS in this study, suggesting that nutrient stress could trigger increased heme synthesis. Nutrient stress is a characteristic feature of the tumor microenvironment, and earlier studies have shown that BC cells (MCF7) grown under glucose deprivation produced higher levels of PpIX than those cultured under standard conditions [66,303]. Furthermore, co-treatment with glycolytic inhibitors and 5-ALA reduced intracellular PpIX levels [304]. This has been attributed to the inactivation of ABC transporters induced by ATP depletion, which in turn decreased the flux of precursors into the cell [303,305]. Interestingly, other studies have reported an increase in cellular PpIX accumulation with the inhibition of ABC transporters [306,307]. On the other hand, a combined treatment with 5-ALA-PDT and dichloroacetate, an inhibitor of pyruvate dehydrogenase, showed improved efficacy in BC (MCF 7) cells [308].

In another study, an 18 h glucose deprivation prior to PDT reduced intracellular glutathione levels and increased the cytotoxicity of PDT [309]. A similar increase in PDT efficacy was observed when inhibitors of glutathione synthesis (buthionine-sulfoximine) or its regeneration (1,3-bis-(2-chlorethyl)-1-nitrosourea) were used for co-treatment with PDT [310]. Changes in the availability of glycolytic substrates affect NADPH availability in the cells. NADPH is a critical agent involved in the anti-oxidative defense mechanisms of the cell. As mentioned, PDT relies on increased ROS in the cancer cells to kill them. PDT-based ROS generation under conditions of impaired glucose and glutathione metabolism results in much higher intracellular ROS levels, contributing to increased efficacy. In line with this, ROS scavengers were shown to protect the cells from ALA-PDT-induced damage [311].

A recent study showed that the metabolic reprogramming toward aerobic glycolysis in cancer cells contributes to the development of resistance to 5-ALA-PDT. Further, they showed that treatment with metformin reduced aerobic glycolysis and increased OXPHOS in squamous cell carcinoma cells, and improved the cytotoxic effect of PDT by increasing PpIX production, ROS generation, and AMPK expression, and inhibiting the AKT/mTOR pathway [312]. In another study, combining glycolysis inhibitor 2-DG with 5-ALA induced an enhanced accumulation of PpIX in HepG2 liver cancer cells [313], contributing to improved treatment efficacy. Further, treatment with 2-DG and 3-bromopyruvate (3-BP) synergistically improved the efficacy of PDT in BC cells [314].

Oncogenic transformation has been reported to upregulate glycolytic enzymes and contribute to the increased exogenous ALA-treatment-induced PpIX accumulation in cells [315]. The increased PpIX accumulation in cancer cells is often attributed to their lower ferrochelatase (FECH) levels. FECH is the terminal enzyme in the heme synthesis pathway that catalyzes the chelation of PpIX with the ferrous ion to generate heme. Lower FECH levels have shown a significant association with the cancer grade, the TNM stage, unfavorable prognosis, and impaired immune cell infiltration in clear cell renal cell carcinoma [316]. However, oncogenic Ras/MEK has been shown to increase the conversion of PpIX to heme by increasing HIF-1α expression and thereby promoting FECH activity [317]. Inhibiting MEK decreased HIF-1α expression and FECH activity and increased 5-ALA-induced PpIX accumulation in cancers [317,318]. Oncogenic Ras/MEK signaling-induced HIF-1α expression has also been implicated in increasing glycolytic flux and driving cancer progression [319,320]. Although 5-ALA-induced PpIX accumulation in cancer cells is a well-documented and well-studied concept, our knowledge of the underlying mechanisms remains largely vague. The inter-dependencies between oncogenic transformation, glycolytic flux, and heme metabolism should be further studied to identify the optimal targeting strategy to enhance 5-ALA-induced PpIX accumulation in cancer cells and to improve the efficacies of PDT and PD.

## 6. Conclusions

The metabolic signatures of tumor cells are different from normal cells, which allows the tumor cells to adapt to the increased energy and metabolite demands [321,322]. Several signaling mechanisms that regulate or hijack the canonical reactions in glucose metabolism have been identified; however, there is no universal mechanism underlying the reprogramming of glucose metabolism in all cancers. Elevated glucose metabolism, hypoxia-induced GLUT, LDH-A, and PFKFB3 overexpression, and the AKT- and c-Myc-mediated transcriptional activation of HK2 are observed in most cancer cells, and these metabolic changes may be exploited for developing effective therapeutic approaches. The tumor microenvironmental regulation and immune suppressive effects of metabolic rewiring are also crucial players in cancer cell progression, survival, and resistance. Apart from glycolytic modulation, mitochondrial dysfunction, elevated ROS production, and dysregulated TCA, cycle enzymes also participate in oncogenic signaling and tumor progression, which were not discussed in this review, and are elegantly reviewed elsewhere [323,324].

Elevated glucose metabolism and nutrient uptake have been exploited for tumor diagnosis in the clinic, and F-18 fluoro-2-deoxyglucose PET is widely used in the clinical staging of cancers [325,326]. However, glycolysis inhibition has not been exploited to its full potential in cancer therapy and it has not translated into the clinic. The inhibition of glycolysis may have undesirable consequences as normal cells also use the same glycolytic enzymes, and it is hence necessary to identify the enzymes or enzyme isoforms that are specifically upregulated or preferentially used by cancer cells. Additionally, several of the glycolysis inhibitors that have been developed for clinical use have been shown to have off-target effects, killing non-cancerous cells [327]. In addition, though the inhibition of glycolysis might inhibit cancer cell proliferation, cancer cells may adapt by upregulating OXPHOS or glutaminolysis, which could result in the development of resistance to therapy, in addition to co-morbidities such as cachexia in patients. This rewiring of metabolic pathways also poses challenges to precision therapies. In fact, previous studies have highlighted the dispensability of the Warburg effect in cancer cell growth and have shown that a complete disruption of glycolysis would require the deletion of both LDHA and LDHB genes. The study evaluated a potent LDH A/B dual inhibitor GNE-140 and demonstrated that the “glycolytic Warburg” phenotype of tumor cells depends on both LDH A and LDH B expression, and is a dispensable phenotype that can be replaced by OXPHOS [328]. This emphasizes the importance of evaluating the simultaneous inhibition of glycolysis and OXPHOS as a therapeutic strategy. In cell lines that are innately resistant to GNE-140 as they predominantly use OXPHOS, the inhibition of OXPHOS sensitized the cells to GNE-140 treatment [329]. Understanding the pathways that contribute to resistance in glycolysis targeting drugs is hence crucial [245,329,330,331].

Despite early indications, glycolytic inhibitors such as 2-DG have failed in the clinical setting due to their limited effects as a monotherapy agent. In addition, it might be necessary to combine glycolytic inhibitors with other agents that target alternate pathways that are activated in response to glycolytic inhibition. Studies have shown promising effects when metformin (that inhibits mitochondrial complex 1) was used along with glycolytic inhibitors. The simultaneous targeting of MCT1 inhibitor AZD3965 and the mitochondrial respiratory complex 1 inhibitor IACS01759 is a more clinically relevant strategy compared with targeting only one pathway in B-cell lymphomas [332]. Using glycolytic inhibitors as adjuvant therapy with already approved conventional therapies, including chemotherapy, radiotherapy, immunotherapy, and photodynamic therapy, is an attractive strategy, and several early studies have shown promising results. Dietary interventions and lifestyle changes also affect the metabolic landscape of cancer cells, and studies should address this issue in detail to emphasize their clinical implications.

Though metabolic rewiring unquestionably affects cancer cell proliferation, the translation of metabolic reprogramming to the clinic must overcome several hurdles [333]. In-depth analytical and extensive pre-clinical studies should identify targetable metabolic enzymes/enzyme isoforms that are efficacious in different tumor types with minimal toxicity to normal cells. Another major challenge in the clinical development of cancer therapeutics is the need to identify patient groups that would benefit from the therapy. Dependency on metabolic pathways is not universal for all cancer types, which makes it all the more important to identify appropriate patient groups to avoid unwanted adverse effects and toxicity. Technical hurdles in measuring metabolism in vivo also add more complexity. It is essential to integrate genetic and metabolic biomarkers and tissue information to optimize the criteria for patient selection. An interesting study published recently showed the possibility of using glycolytic enzymes as a surrogate for glycolytic activity in cancer cells, using liquid biopsy. Cells expressing high levels of HK2 were identified in both cytokeratin (CK)-positive and CK-negative CTC populations isolated from lung adenocarcinoma patients [334]. The possibility of monitoring the glycolytic metabolic rewiring in real-time using liquid biopsy samples and downstream single-cell level molecular, genomic, and metabolic studies is likely to create a paradigm shift in the clinical utility of glycolytic modulations in cancer therapy. Well-designed clinical trials that unravel the metabolic dependencies of different cancer types and their association with genetic and histopathological information might answer several questions on the complex and sophisticated nature of cancer metabolism. Metabolomic studies, with powerful technological backing, would take the lead in the early identification of cancer risk factors and aid in improving cancer therapy and cancer screening, diagnosis, and therapy monitoring in the coming years.

## Figures and Tables

**Figure 1 ijms-24-02606-f001:**
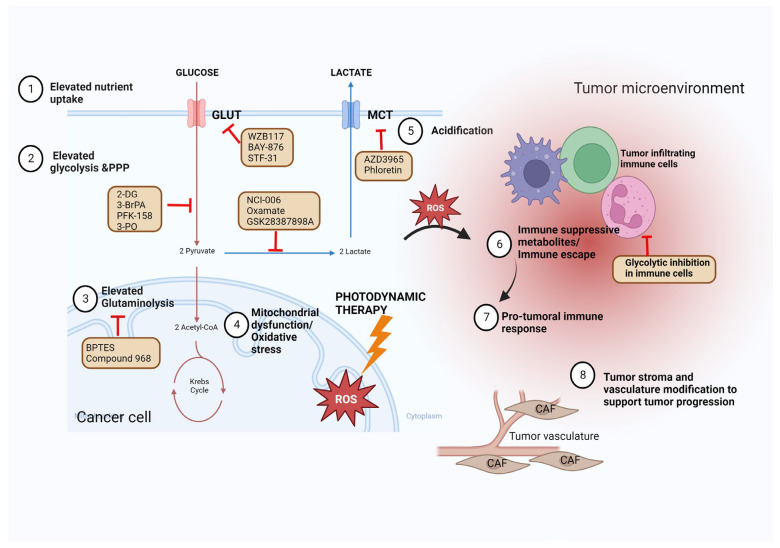
Targeting glycolysis to improve cancer therapy. The cancer cells show enhanced dependency on glycolysis that may be targeted to improve the treatment efficacy of conventional cancer therapy modalities, including chemotherapy, radiotherapy, immunotherapy, hormonal therapy, and photodynamic therapy. Glycolysis metabolism can be potentially targeted by limiting glucose uptake (targeting glucose transporters), targeting glycolysis enzymes, targeting glutaminolysis, targeting lactate synthesis, targeting MCT, or targeting mitochondrial complexes. The increased glycolysis in the cancer cells increase the release of lactic acid to the tumor microenvironment, acidifying it and making it pro-cancer and immunosuppressive. The tumor-infiltrating immune cells also show a similar shift toward glucose metabolism increasing the competition for glucose in the tumor microenvironment. Modulating glycolysis in the immune cells can potentially improve immune therapy. This figure was created using the Biorender app.

**Figure 2 ijms-24-02606-f002:**
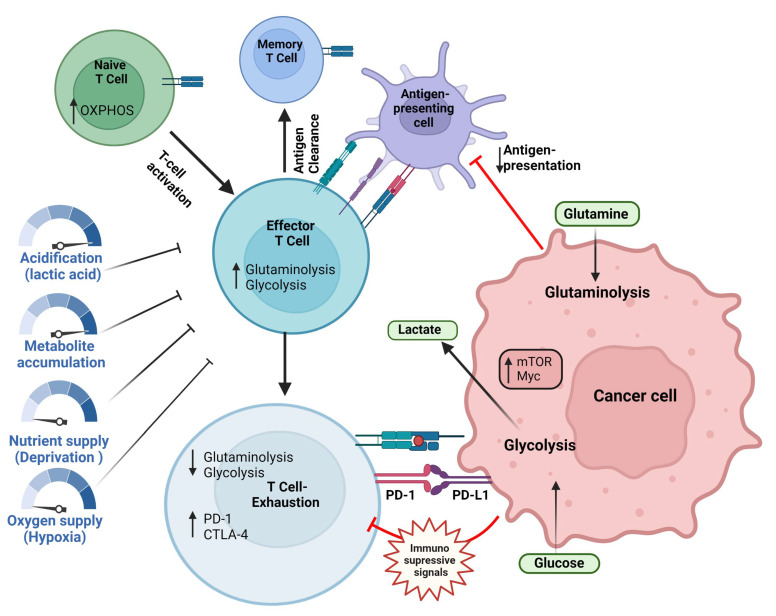
Naïve T-cells relay on oxidative metabolism. After activation, the effector T-cells increase glycolysis to support their function. On antigen clearance, the effector T-cells enter a memory state. On antigen persistence, as with long-term tumor elimination, inhibitory receptors such as PD-1 and CTLA4 reprogram the T-cell metabolism leading to metabolic impairments. Exhausted T-cells show reduced glycolysis and glutaminolysis and dependence of fatty acid oxidation. The low levels of oxygen, high levels of lactate, and the high competition for glucose potentially contribute to T-cell dysfunction in the tumor microenvironment. The image was created using Biorender app.

**Figure 3 ijms-24-02606-f003:**
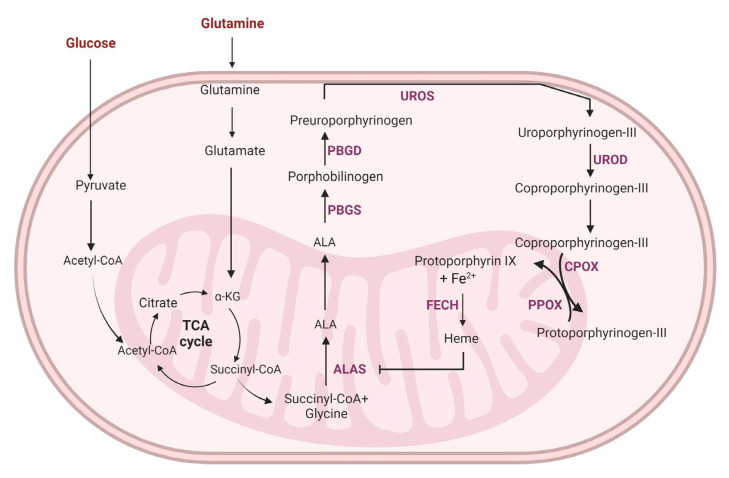
The heme biosynthesis is connected with glucose and glutamine metabolism. Enhanced glycolysis and glutaminolysis in the cancer cell might contribute to the elevated rate of heme synthesis and the upregulation of several heme biosynthesis pathway enzymes in the cancers. Addition of exogenous 5-ALA bypasses the feedback inhibition of ALAS and increase the accumulation of protoporphyrin IX (PpIX) in the cancer cells. The elevated accumulation of PpIX is exploited for fluorescence-guided detection (photodynamic diagnosis) of cancers. Further, irradiating PpIX-accumulating cancer cells with light generated singlet oxygen and ROS that kills the cancer cell, with minimal damage to the surrounding tissue (photodynamic therapy; PDT). The interdependency of glucose metabolism and the heme synthesis pathways suggest that targeting glycolysis could enable the modulation of PpIX accumulation in the cancer cells. The image was created using Biorender app.

## Data Availability

Not applicable.

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
