# Peer review of "Modulating Glycolysis to Improve Cancer Therapy"

_ijms, 2023, doi:10.3390/ijms24032606_

Round 1

Reviewer 1 Report

The manuscript Chelakkot et al. submitted to IJMS entitled: “Modulating glycolysis to improve cancer therapy” " appears to be a very comprehensive review with more than 300 references focusing on the effect of glycolysis, known as the Warburg effect, on tumors and tumor therapies.

However, in the first sentence of the chapter introduction, the authors mentioned that “cancer promotes growth, ..., proliferation”, which describes the same thing, I guess.

In the second sentence of the chapter introduction, the authors describe "increased glucose uptake and fermentation of glucose to lactose." At this point and at the beginning of this work, the authors describe that a c6 body (glucose) is fermented into a disaccharide lactose (D-galactose and D-glucose). This is absolutely wrong, the authors mean lactic acid, which is a c3 body!

 In the 5th sentence of Chapter 2.2, the authors described that HK1 and HK2 are mitochondrial enzymes, but both enzymes are cytosolic enzymes that are located on the outer mitochondrial membrane!   The point is that glycolysis takes place in the cytoplasm and not in the mitochondria. And at this point, as an expert in this field, you should ask yourself why some enzymes in this pathway should be mitochondrial. The answer can be found in any biochemistry textbook.

In chapter 2.3, sentence 7, the authors wrote "oncogenes like wnt..", but not what kind of wnt (wnt3a, wnt 9 or what) they want to describe as an oncogene.

Scientists, who should be experts in this field of research, must not make such mistakes. 

Author Response

The manuscript Chelakkot et al. submitted to IJMS entitled: “Modulating glycolysis to improve cancer therapy” " appears to be a very comprehensive review with more than 300 references focusing on the effect of glycolysis, known as the Warburg effect, on tumors and tumor therapies.

However, in the first sentence of the chapter introduction, the authors mentioned that “cancer promotes growth, ..., proliferation”, which describes the same thing, I guess.

Response: We have modified the sentence to address this redundancy. Please see the revised sentence below.

Cancer cells reprogram their metabolism to promote growth, metastasis, invasion and  survival.

In the second sentence of the chapter introduction, the authors describe "increased glucose uptake and fermentation of glucose to lactose." At this point and at the beginning of this work, the authors describe that a c6 body (glucose) is fermented into a disaccharide lactose (D-galactose and D-glucose). This is absolutely wrong, the authors mean lactic acid, which is a c3 body!

Response: Thanks for pointing this out. This was a mistake on our part. It should have been ‘lactic acid’ as the reviewer pointed out and not ‘lactose’. We have corrected this in the revised manuscript.

In the 5th sentence of Chapter 2.2, the authors described that HK1 and HK2 are mitochondrial enzymes, but both enzymes are cytosolic enzymes that are located on the outer mitochondrial membrane! The point is that glycolysis takes place in the cytoplasm and not in the mitochondria. And at this point, as an expert in this field, you should ask yourself why some enzymes in this pathway should be mitochondrial. The answer can be found in any biochemistry textbook.

Response: We agree with the reviewer on the localization of HK1 and HK2. We have revised the text as shown below.

Four HK isoforms, HK1-4, with different cellular distributions and glucose affinity have been identified. HK1 and HK2 are located on the outer mitochondrial membrane and are associated with AKT-mediated cell survival.

In chapter 2.3, sentence 7, the authors wrote "oncogenes like wnt..", but not what kind of wnt (wnt3a, wnt 9 or what) they want to describe as an oncogene.

Response: Here, we intended to highlight that the cancer cells have altered signaling pathways, activated by the DNA repair pathways, which contribute to their prolonged survival and apoptosis resistance. Since we are mentioning more than one pathway that might be activated, we did not specify which of the Wnt ligands are involved. However, based on the reviewers’ comment, we have modified the text slightly (see below) to highlight that the DNA repair pathways contribute to the prolonged survival.

The DNA repair pathways in reprogrammed cancer cells induce the activation of several pro-tumorigenic signaling pathways, including Wnt, PI3K/AKT, NF-κB, and MAPK, triggering prolonged cancer cell survival and apoptosis resistance.

We thank the reviewers once again for their comments and suggestions for improving the manuscript. We hope that we have addressed the reviewers’ comments satisfactorily.

Reviewer 2 Report

Kyoung Song and colleagues have developed a large comprehensive revue entitled “Modulating glycolysis to improve cancer Therapy”

They provided a large survey of the literature on a very important topic. This review is descriptive and will have benefit of being more critical on the presentation of the literature.

The major problem of the glycolysis field is that all the drugs that have been developed by the big Pharma over the last 20 years are non-specific. Some could beautifully arrest tumour growth like LDHA inhibitors but their toxicity via off target effects has prevented their exploitation in the clinic. The same could be said for 2-deoxyglucose or even worse for the 3-Bromopyruvate acid 3BrPA classified as a glycolytic inhibitor. In fact 3BrPA kills cells by energy depletion. In 30 minutes of treatment, ATP drops to extreme low levels by inhibiting as well pyruvate dehydrogenase. 

What I find missing in this review is the fact that Warburg effect is dispensable for tumour growth. Reviewer’s team has demonstrated that full genetic disruption of lactic acid formation rewire tumour cells to OXPHOS with lower growth rate but no cell death.

Complete disruption of lactate formation requires deletion of LDHA and LDHB genes.

This is has fully been verified by the inhibitor GNE140 developed by Genentech that inhibit both LDHA and LDHB mimicking combined genetic deletion of LDHA and LDHB.   

Authors might want to cite the key articles with reference to GNE140 (ref 1,2) and a recent review initiated for the centennial of Otto Warburg discovery: (ref 3)

(1) Ždralević M, Brand A, Di Ianni L, Dettmer K, Reinders J, Singer K, Peter K, Schnell A, Bruss C, Decking SM, Koehl G, Felipe-Abrio B, Durivault J, Bayer P, Evangelista M, O'Brien T, Oefner PJ, Renner K, Pouysségur J, Kreutz M. Double genetic disruption of lactate dehydrogenases A and B is required to ablate the "Warburg effect" restricting tumor growth to oxidative metabolism. J Biol Chem. 2018 Oct 12;293(41):15947-15961.

(ref 2) Boudreau A, Purkey HE, Hitz A, Robarge K, Peterson D, Labadie S, Kwong M, Hong R, Gao M, Del Nagro C, Pusapati R, Ma S, Salphati L, Pang J, Zhou A, Lai T, Li Y, Chen Z, Wei B, Yen I, Sideris S, McCleland M, Firestein R, Corson L, Vanderbilt A, Williams S, Daemen A, Belvin M, Eigenbrot C, Jackson PK, Malek S, Hatzivassiliou G, Sampath D, Evangelista M, O'Brien T. Metabolic plasticity underpins innate and acquired resistance to LDHA inhibition. Nat Chem Biol. 2016 Oct;12(10):779-86.

(ref 3) Pouysségur J, Marchiq I, Parks SK, Durivault J, Ždralević M, Vucetic M. 'Warburg effect' controls tumor growth, bacterial, viral infections and immunity - Genetic deconstruction and therapeutic perspectives. Semin Cancer Biol. 2022 Nov;86(Pt 2):334-346.

Author Response

We thank the reviewers for carefully reviewing our manuscript and for providing valuable suggestions and comments. We have carefully revised the manuscript based on the comments and suggestions provided by the reviewers. We have provided a point-by-point response to the reviewers’ comments below. We have indicated our responses using blue font.

Response to Reviewer 2’s comments

Kyoung Song and colleagues have developed a large comprehensive revue entitled “Modulating glycolysis to improve cancer Therapy”

They provided a large survey of the literature on a very important topic. This review is descriptive and will have benefit of being more critical on the presentation of the literature.

Response: Thank you very much for the positive feedback.

The major problem of the glycolysis field is that all the drugs that have been developed by the big Pharma over the last 20 years are non-specific. Some could beautifully arrest tumour growth like LDHA inhibitors but their toxicity via off target effects has prevented their exploitation in the clinic. The same could be said for 2-deoxyglucose or even worse for the 3-Bromopyruvate acid 3BrPA classified as a glycolytic inhibitor. In fact 3BrPA kills cells by energy depletion. In 30 minutes of treatment, ATP drops to extreme low levels by inhibiting as well pyruvate dehydrogenase.

Response: We agree with the reviewer on this point, and have discussed this point in the manuscript as well. To further highlight the off-target effect and toxicity of the various glycolysis inhibitors, we have added a sentence regarding this (please see below) and have cited relevant literature in the conclusion.

Additionally, several of the glycolysis inhibitors that have been developed for clinical use have been shown to have off-target effect, killing non-cancerous cells.

What I find missing in this review is the fact that Warburg effect is dispensable for tumour growth. Reviewer’s team has demonstrated that full genetic disruption of lactic acid formation rewire tumour cells to OXPHOS with lower growth rate but no cell death. Complete disruption of lactate formation requires deletion of LDHA and LDHB genes.

Response: We agree with this point, and have highlighted that cancer cells can adapt to glycolysis inhibition by upregulating other pathways. We thank the reviewer for suggesting additional references for the manuscript. Following the reviewer’s suggestion, we have cited the suggested articles in the concluding section (please see below).

This rewiring of metabolic pathways also pose challenge to precision therapies. In fact, previous studies have highlighted the dispensability of the Warburg effect in cancer cell growth and have shown that a complete disruption of glycolysis would require the deletion of both LDHA and LDHB genes. The study evaluated a potent LDH A/B dual inhibitor GNE-140 and demonstrated that “glycolytic Warburg” phenotype of tumor cells depends on both LDH A and LDH B expression, and is a dispensable phenotype that can be replaced by OXPHOS [328]. This emphasizes the importance of evaluating simultaneous inhibition of glycolysis and OXPHOS as a therapeutic strategy. In cell lines that are innately resistant to GNE-140 as they predominantly use OXPHOS, inhibition of OXPHOS sensitized the cells to GNE-140 treatment [329]. Understanding the pathways that contribute to resistance in glycolysis targeting drugs is hence crucial. [245,329,330,331].

We thank the reviewers once again for their comments and suggestions for improving the manuscript. We hope that we have addressed the reviewers’ comments satisfactorily.